

Comment on: "A review of the complementary principle of evaporation: From the original linear
relationship to generalized nonlinear functions" by S. Han and F. Tian
Richard D. Crago[1], Jozsef Szilagyi[2], Russell Qualls[3]
[1] Department of Civil and Environmental Engineering, Bucknell University, Lewisburg, PA,
USA
[2] Department of Hydraulic and Water Resources Engineering, Budapest University of
Technology and Economics, Budapest, Hungary; also at School of Natural Resources, University
of Nebraska, Lincoln, Lincoln, NE, USA
[3] Department of Biological Engineering, University of Idaho, Moscow, ID, USA
**Abstract**
The paper by Han and Tian reviews the history of developments in the complementary
relationship (CR) between actual and potential evaporation and introduces the generalized
complementary principle (GCP) developed by the authors. This comment assesses whether the
GCP: 1) Can give reasonable results from a wide range of surfaces worldwide; 2) is supported by
experimental data that verify the three-stages of evaporation implicit in the GCP, particularly in
the wet-surface limit; 3) has been proven to be correct by the authors in a previous paper; and 4)
is supported by model studies showing that wet surfaces occur predominantly during periods of



large-scale moisture convergence. The assessment finds that arguments in favor of the GCP
deserve to be taken seriously, but ultimately remain unconvincing.

**1. Introduction**
Han and Tian (2020) (hereafter HT20) provide important insights into the growing body of
literature regarding the Complementary Relationship (CR) of evaporation, and serves well as an
accessible review of the literature. The sigmoid formulation (their equation 13), a key feature of
their Generalized Complementary Principle (GCP) (Han and Tian, 2018; hereafter HT18) is
presented and defended in their paper.
Two of the present authors (Szilagyi and Crago, 2019, hereafter SC19) wrote an earlier comment
critiquing the sigmoid function for violating established physical principles (see also the reply by
Han and Tian, 2019a). After further consideration, the present authors recognize that the
Priestley and Taylor (1972) line at $x_H = E_{rad}/E_{Pen}=1/\alpha=1/1.26$  that appears in HT20 (their Figure
3), could be intended by HT18 and HT20 to mark a reference point on the graph, rather than to
establish a limiting value that cannot be crossed. Unless otherwise noted, all notation herein
follows that of HT20—see also Tables I and II for notation and variable names. Also, the role of
a related (but different) adjustable parameter (also named α) seems to be used in the formulation
primarily to adjust the shape of the sigmoid curve, rather than to set a limit on wet surface
evaporation.





Table I Variables used

| $b$ | A GCP model parameter that adjusts the shape of the sigmoid function |
|---|---|
| $E$ | Actual regional evaporation rate |
| $E_{aero}$ | The second term of Penman's (1948) equation, related to the drying power of the air. |
| $E^{max}_{MT}$ | Hypothetical maximum value of E that would occur from a wet patch in an otherwise completely desiccated region |
| $E_{Pen}$ | Evaporation rate from Penman's (1948) equation |
| $E_{PT}$ | $\alpha E_{rad}$ proposed by Priestley and Taylor (1972) for a wet regional surface with minimal advection |
| $E_{rad}$ | The first term of Penman's (1948) equation, equivalent to the equilibrium evaporation rate of Slatyer and McIlroy (1961) |
| $E^{Tws}_{PT}$ | Value of EPT found if the slope of the saturation vapor pressure curve is estimated at the wet surface temperature, $T_{ws}$ (see Szilagyi et al., 2016) |
| $f(E_{rad}/E_{Pen})$ | A hypothesized function of $E_{rad}/E_{Pen}$ |
| $x_H$ | $E_{rad} / E_{Pen}$ |
| $x_m$ | $E^{Tws}_{PT} / E^{max}_{MT}$ the value of $E^{Tws}_{PT} / E_{Pen}$ at which $E$ goes to zero in the rescaled CR (Crago et al, 2016) |
| $x_{max}$ | Parameter that sets the maximum value $x_H$ can reach |
| $x_{min}$ | Parameter that sets the value of $x_H$ at which $y_H \to 0$ |
| $y_H$ | $E / E_{Pen}$ |
| $\alpha$ | The Priestley & Taylor (1972) parameter |






Table II. Abbreviations

| BC4 | Boundary condition 4: $\mathrm{d}(E/E_{\mathrm{pen}})/\mathrm{d}(E_{\mathrm{rad}}/E_{\mathrm{pen}}) = \mathrm{d}y_H/\mathrm{d}x_H \to 0$ as as $y_H \to 1$ |
|------|-------------------------------------------------------------------------------------------------------------------------|
| CR | Complementary Relationship (between actual and potential evaporation) proposed by Bouchet (1963) |
| GCP | Han and Tian's (2020) Generalized Complementary Principle |
| HT18 | Han and Tian (2018) |
| HT20 | Han and Tian (2020) |
| SC19 | Szilagyi and Crago (2019) |


The most controversial feature of the sigmoid function is the slope of the curve at the wet-surface
limit. Namely, it requires that $\mathrm{d}(E/E_{\mathrm{pen}})/\mathrm{d}(E_{\mathrm{rad}}/E_{\mathrm{pen}}) = \mathrm{d}y_H/\mathrm{d}x_H \to 0$ as as $y_H \to 1$ (hereafter, this
boundary condition will be denoted "BC4"). That is, rather than a complementary relationship,
BC4 requires that $E$ and $E_{\mathrm{Pen}}$ are equal and that $E$ exactly follows any variability by $E_{\mathrm{Pen}}$ in the
wet surface limit.
BC4 deserves careful attention. A major purpose of this comment is to show that there are some
indications such behavior can occur, but when it does it is a consequence of large-scale processes
that disconnect the regional land surface from the overlying atmosphere, thus violating the basic
assumptions behind the CR (namely, that atmospheric and surface conditions are tightly linked
through surface fluxes). In light of this, corrections to the CR attempting to account for these
exceptional cases will inevitably result in a formulation that does not accurately represent
ordinary (minimally-advective) conditions.





This comment will consider the evidence for the following four claims made by HT18 and HT20
in support of the sigmoid function and BC4: First, that the function works reasonably well to
model evaporation from sites around the world; second, that data from these sites support a
three-stage evaporation process and BC4, both of which are required by the sigmoid function;
third, that HT2018 have provided a rigorous proof of the boundary conditions underlying the
formulation; and fourth, that a partial explanation of BC4 has been provided by the study of
Lintner et al. (2015).
**2.  Claim regarding modeling results**
First, it is clear that the sigmoid function has been used successfully to model evaporation from
flux stations around the world (see HT18). It is quite a flexible formulation that can match a wide
range of data patterns on an ($x_H$, $y_H$) graph. Calibrated values of α and $b$ published in HT18 (their
Table 5) range from about 1.01 to 1.49 and from 0.59 to 17, respectively. Figure 1 shows the
sigmoid function for the four combinations of these extreme parameter values (with $x_{min}$=0 and
$x_{max}$=1). These show the wide range of possible curve shapes; allowing $x_{min}$ and $x_{max}$ to take other
fixed values further increases the flexibility. Such an equation is likely to fit many datasets well,
if tuning is permitted. Of course, any CR formulation must ultimately work well without
requiring local calibration of parameters.






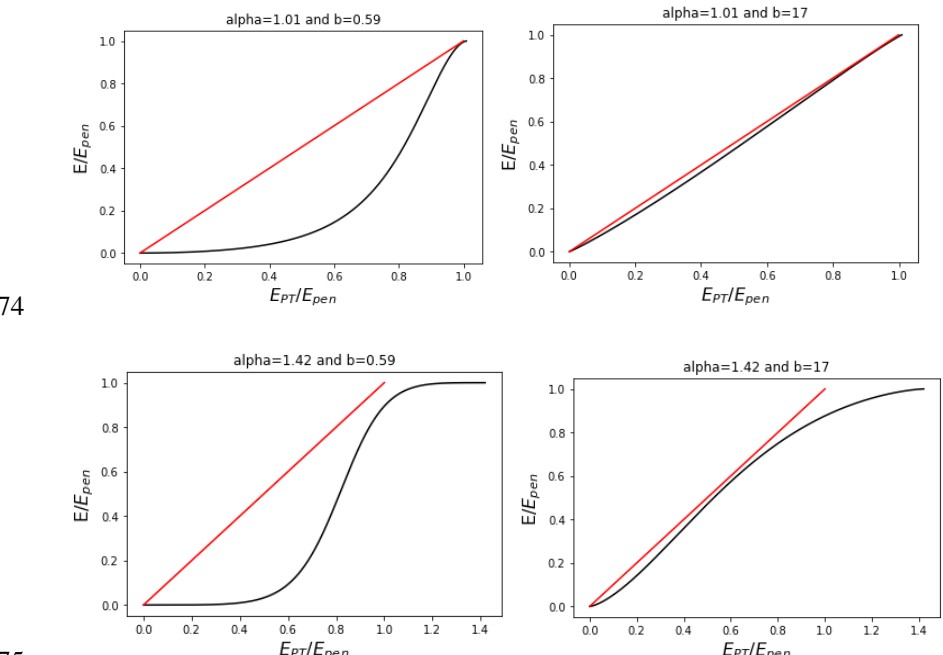


Figure 1. The sigmoid function (black curves) and the Priestley-Taylor line (α=1.26, straight line

in red) for the most extreme parameter values documented in HT18. The scales of the horizontal

axes differ.


**3. Claim regarding empirical support for three evaporation stages and for BC4**

Second, there does seem to be some empirical support for different slopes at different positions

on $(x_H, y_H)$ graphs (HT18, their Table 3). However, the curve proposed by Brutsaert (2015) also

proposes a shallow slope for small $y_H$,(stage 1) a steep slope in the middle (stage 3), and a less

steep slope near $y_H=1$ (stage 3). Similar behavior is also possible with the rescaled models of the

present authors. The stage 3 slopes at large $y_H$ values (HT18, Table 3) would be near zero

according to BC4, but are generally near 1 instead. HT18 directly address BC4 with data in their

Figure 6, which plots empirical data along with red curves resulting from the sigmoid function





relating $E/E_{\text{PT}}$ to $E/E_{\text{Pen}}$. The sigmoid function curves show $E/E_{\text{PT}}$ increasing as $E/E_{\text{Pen}}$ increases,
until $E/E_{\text{PT}}$ reaches a peak and then begins to decrease with further increases in $E/E_{\text{Pen}}$.
Correlational evidence for this downturn is given by HT18, but the actual data plotted do not
visibly follow the downturn in $E/E_{\text{PT}}$ in either panel of Figure 6; the dramatic downturn in the red
curve Figure 6(a) (the left panel) certainly is not matched by the data.
**4. Claim regarding the derivation by HT18**
Third, the derivation by HT18 is inconclusive. The derivation begins [HT18, their Eq. (8)]:
$E = (E_{\text{pen}}) * f(E_{\text{rad}} / E_{\text{pen}})$, where $E_{\text{pen}} = E_{\text{rad}} + E_{\text{aero}}$         (1)

where $f$ is a function of $(E_{\text{rad}}/E_{\text{pen}})$. Partial derivatives of $E$ were taken from Eq. (1) with respect
to $E_{\text{rad}}$ and $E_{\text{aero}}$. Further manipulations of these derivatives resulted in the four boundary
conditions corresponding to the sigmoid curve (HT18). The function $f(E_{\text{rad}}/E_{\text{pen}})$ in Eq. (1) could
include constants or parameters (for instance $\alpha$, $x_{\text{min}}$, and $x_{\text{max}}$), whose "correct" values can be
found by calibration, after which they must be treated as constants. This means that, once the
parameters are determined, the shape of $f(E_{\text{rad}}/E_{\text{pen}})$ is also determined.

Unfortunately, this leads to two problems. First, the present authors' work with the "rescaled" CR
(Crago et al., 2017, Szilagyi et al., 2017, Crago and Qualls, 2018) gives evidence that the
variable $x_{\text{m}} = E^{\text{Tws}}_{\text{PT}} / E^{\text{max}}_{\text{MT}}$, ($x_{\text{m}}$ is our own notation) related to the value of $E^{\text{Tws}}_{\text{PT}} /E_{\text{Pen}}$ at
which $E$ goes to zero, is in fact a variable, not a constant. It must be calculated for each
individual data point, and it results in a significant re-arrangement of the data. It could have been
included in Eq. (1) by writing Eq. (1) as: $y_{\text{H}} = f(x_{\text{H}}, x_{\text{m}})$. By taking derivatives without including
the impact that a variable $x_{\text{m}}$ might have, HT18 assumed from the beginning that $E/E_{\text{pen}}$ does not





vary with $x_m$, so a variable $x_m$ boundary condition could not possibly arise from this derivation.
On the other hand, if $x_m$ is in fact a significant variable (as the papers cited above suggest), it
could impact the entire derivation, but particularly the two dry-limit boundary conditions.

The parameter $x_{max}$ is the maximum value $x_H$ can reach, and is usually taken by HT18 and HT20
to be $1.26^{-1}$, where 1.26 is the commonly-accepted value for the Priestley and Taylor parameter
$\alpha$. To prove that $dy_H/dx_H \rightarrow 0$ as $y_H \rightarrow 1$ (the most controversial finding of the derivation), HT18
had to show that $\partial x_{max}/\partial E_{rad}$ evaluated at $y=1$ cannot be 0 (see the paragraph starting at the
bottom of page 5054 and ending at the top of page 5055 of HT18). But if Eq. (1) is true, $x_{max}$ has
to be treated as a constant, so the partial derivative must be 0. It is impossible for $x_{max}$ to be a
constant for the purpose of taking derivatives of Eq. (1), but a variable when evaluating
$\partial x_{max}/\partial E_{rad}$. Thus, there is a logical inconsistency hidden in this derivation. SC19 showed that, if
the Priestley-Taylor $\alpha$ (equivalent here to $1/x_{max}$) is actually a constant, HT18's derivation does
not result in a specific required value for $dy_H/dx_H$ at $y=1$. Thus, the boundary condition
$dy_H/dx_H \rightarrow 0$ as $y_H \rightarrow 1$ does not follow from (1).

To sum up consideration of the derivation, three of the four boundary conditions (slope and
intercept at the point where $y_H \rightarrow 0$, and slope as $y_H \rightarrow 1$) are doubtful due to the assumptions made
when (1) was used as the definition of $E$.

**5.  Claim regarding support from the modeling study of Lintner (2015)**
HT18 cite the modeling results of Lintner et al. (2015) in support of BC4. This study used a
steady-state model that captured the key physical processes affecting evaporation. Model results



show decreases in both $E_{\text{Pen}}$ and $E$ as soil moisture approaches saturation, similar to the behavior
required by BC4. According to Lintner (see also HT18), large-scale horizontal moisture
convergence decreases $E_{\text{Pen}}$ by increasing atmospheric humidity, and at the same time it
increases precipitation and thus soil moisture content. Near the wet limit, water availability
matters less than $E_{\text{Pen}}$ in determining $E$, so $E$ and $E_{\text{Pen}}$ decrease at the same rate. Thus, at the point
of saturation, $E=E_{\text{Pen}}$, and $\mathrm{d}(E/E_{\text{Pen}})/\mathrm{d}(E_{\text{PT}}/E_{\text{Pen}}) = 0$, apparently satisfying BC4.
But note that the normal (i.e., minimal moisture convergence, divergence, or advection) behavior
for a wet surface is $E=E_{\text{Pen}}=E_{\text{PT}}$ (e.g., Brutsaert, 2015). The only way to get BC4-type behavior is
to impose a large-scale process that causes $E_{\text{Pen}}$ to differ from this value. That is, BC4 is not
describing the drying process and the CR at all; rather, it is describing what happens when the
CR simply does not apply. The scenario described by Lintner et al. (2015) requires a clear
disconnect between the land surface processes and the overlying atmospheric conditions,
violating the central assumption of the CR (e.g., Brutsaert, 1982, 2005).
It need not be the case that nearly-saturated surfaces coincide with moisture convergence outside
of steady-state models. Nearly-saturated surface conditions can exist under a range of large-scale
patterns, including positive, negative or negligible moisture convergence or advection. This is
the case because soil moisture content varies at larger time scales than most other components of
the surface water and energy budgets (e.g., Sellers et al, 1992), so nearly-saturated surface
conditions can persist after a period of moisture convergence has ended. Furthermore, saturated
surfaces can occur from other processes, such as thunderstorms driven by surface heating.
**6. Conclusions**





HT18 and HT20 have martialed several empirical and theoretical arguments in support of their
proposed sigmoid formulation of the CR. The range of arguments and data sources used is
impressive, and the present authors only recently recognized the specific nature and the impact
of this challenge to other CR formulations. There is little doubt that some aspects of their
argument are true, including the ability of their formulation to match numerous experimental
datasets. Nevertheless, the specific boundary conditions leading to the sigmoid function are not
well-supported by empirical data; the derivation of the boundary conditions by HT18 was
inconsistent regarding which model values are constants and which are variables; and the
argument that large-scale processes require adoption of BC4 fails because it essentially makes
the exception (large-scale processes dominating land surface processes in determining near-
surface atmospheric conditions) into the rule, and in doing so, it violates the assumptions of the
CR. The CR should ideally only be used under circumstances where advection is minimal.
Attempts to adjust for other conditions (e.g., Parlange and Katul, 1992) are possible, but should
not over-ride consideration of the basic CR concept. This may require developing specific
conditions for screening data.
There does not seem to be consensus in the research community on any of the boundary
conditions of the CR except for $x_H=1$ when $y_H=1$. The current authors find the evidence for a
variable $x_m$ to be strong. This value can be calculated separately for each data point and it leads
to a rescaling of the $x_H$-axis, and a resulting reduction in the scatter of the data points (Crago and
Qualls, 2018).
While the sigmoid formulation is clearly the result of a serious and substantial research program,
the difficulties with it described here are serious enough that we cannot see it as an improvement
over other recent CR formulations.





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
