# Peer review of "Comment on: "A review of the complementary principle of evaporation: From the original linear relationship to generalized nonlinear functions" by S. Han and F. Tian"

_Hydrology and Earth System Sciences, 2020_

## Referee Comment (RC1) · Songjun Han (Referee) · 2 Sep 2020

**Review of hess-2020-310 "Comment on: A review of the complementary principle of evaporation: From the original linear relationship to generalized nonlinear functions by S. Han and F. Tian"**

This is a review for this manuscript on behalf of S. Han and F. Tian.

This manuscript is a comment on the sigmoid generalized complementary (hereafter SGC) principle, which was developed by us in Han et al. (2012) and Han and Tian (2018) (hereafter HT18), and was reviewed in Han and Tian (2020) (hereafter HT20). Thus, this review can also be regarded as our reply to the authors' comment.

We are happy to have the conversations regarding the complementary principle, and would like to thank R. Crago, J. Szilagyi and R. Qualls for their comment. We think this manuscript is worth for publication after considering comments below.

To the best of our understanding, the authors have four claims in the comment: (1) mathematical local calibration, (2) physical assumption of the CR, (3) observational support and (4) theoretical derivation for the boundary conditions. Each of the claim will be listed and commented separately below.

1)  The SGC equation models evaporation with two calibrated parameters (*a* and *b* in HT18), which violates the aim of former CR: "without requiring local calibration".

    Comments: To some extent, we agree with the argument that "any CR formulation must ultimately work well without requiring local calibration of parameters". In fact, calibration-free is the dream of any model development. However, there are two routes in the studies of CR leading to this ultimate objective. One adopts an existing complementary relationship with default parameter(s), and concentrates on properly formulating the potential evaporation Epo and/or apparent potential evaporation Epa, or carefully rescaling the independent variable of an existing CR model. The authors' work with the "rescaled" CR (Crago & Qualls, 2018; Crago et al., 2016; Szilagyi et al., 2017) follows this route. We believe that, if proper formulations of Epa and/or Epo,

and/or appropriate rescaling approaches are carefully conducted on a physical basis, a calibration-free CR evaporation estimation model could be ultimately achieved.

The other route, calibrating parameters for the fitting of observed points and proposing a method to determine the parameters *in priori*, is widely used in evaporation modeling (Monteith, 1965; Shuttleworth, 1993; Yang et al., 2007). Local calibration is just the first step. After the first step of local calibration, we have been working on the priori determinate of the parameters. In our recent published paper (Wang et al., 2020), we found that the parameter *b* changes with the ecosystem type, and used the ecosystem mean b values of 217 sites around the world in the B2017 with little weakening of the evaporation estimation accuracy. We are also working at the characteristics and determination methods of the other parameter *a* (Han et al., 2020).

In our opinion, both methodologies are deserved to be explored.

2) The SGC equation adopts a wet boundary condition (denoted as BC4 in the manuscript) only occurring when advections from outside (Brutsaert & Stricker, 1979) or large-scale synoptic changes (Liu et al., 2011; Shuttleworth et al., 2009) play important roles in determining the near-surface atmospheric variables, which violates the central assumption of the CR: the land surface wetness can be effectively detected from the overlying drying power of air with a constant radiation energy input (e.g., Brutsaert, 1982, 2005).

Comment: We totally agree that our wet boundary condition violates the central assumption of the CR. However, this can be regarded as an extension of original CR principle.

The complementary principle was originally proposed for the evaporation taking place from "a sufficiently large and homogeneous surface" (Brutsaert, 2015), over which the advection effects of heat and water vapor from outside are negligible or changeless (Brutsaert & Stricker, 1979; Han & Tian, 2018b; Morton, 1983). The complementary principle employs an assumption that the land surface wetness can be effectively detected from the drying power of air with a constant radiation energy input (Brutsaert, 1982; Han & Tian, 2018a, 2020). Following this assumption, we expressed $\frac{E}{E_{Pen}}$ as a function of the atmospheric wetness index $\frac{E_{rad}}{E_{Pen}}$, claimed that

"the spirit of the complementary principle is still retained" (Han & Tian, 2018a), and proposed a SGC function.

The boundary condition BC4 and the upper flatness feature of the SGC function require that both $E$ and $E_{aero}$ increase with constant $E_{rad}$ when $E/E_{Pen}$ decreases from the unit (Han & Tian, 2018a, 2019). If $E_{rad}$ is constant, an increase in $E_{aero}$ means an increase in the vapor pressure deficit if the wind speed is changeless. But according to the assumption of the complementary principle, the vapor pressure deficit could only increase if less water was evaporated into the air, which means a decrease in $E$. Thus, BC4 and the upper flatness feature of the growth of $\frac{E}{E_{Pen}}$ upon $\frac{E_{rad}}{E_{Pen}}$ was questioned by considering that it is impossible for $E$ and $E_{aero}$ change in the same direction over a nearly wet surface (Szilagyi & Crago, 2019).

However, over a natural landscape, there are situations that advections from outside (Brutsaert & Stricker, 1979) or large-scale synoptic changes (Lintner et al., 2015; Shuttleworth et al., 2009) play important roles in determining the near-surface atmospheric variables. Then, the land surface and the atmosphere are not necessarily fully coupled. If considering the second type processes, the same direction changes in $E$ and $E_{aero}$ could be understood. For example, the horizontal advection of hot dry air to the wet surface would enhance both $E$ and $E_{aero}$ with constant $E_{rad}$. Thus the growth of $\frac{E}{E_{Pen}}$ upon $\frac{E_{rad}}{E_{Pen}}$ is slower with larger values of $\frac{E_{rad}}{E_{Pen}}$, which is an upper flatness feature.

In our preparing paper (Han et al., 2020), we investigated the relationship between $\frac{E}{E_{Pen}}$ and $\frac{E_{rad}}{E_{Pen}}$ over wet surfaces as the extremes. Although the assumption of the complementary principle does not hold over the land surface with ample and changeless water availability, $\frac{E_{rad}}{E_{Pen}}$ varies significantly due to the advections or the large-scale synoptic changes, and $\frac{E}{E_{Pen}}$ is still highly related with $\frac{E_{rad}}{E_{Pen}}$, and their relationship still can be described be the SGC equation. Especially, the growth of $\frac{E}{E_{Pen}}$ upon $\frac{E_{rad}}{E_{Pen}}$ (as well as $E$ upon $E_{rad}$) exhibits nonlinear characteristics with slowing down growth rate for large values of $\frac{E_{rad}}{E_{Pen}}$. The results imply that $\frac{E}{E_{Pen}}$ can be expressed as a function of $\frac{E_{rad}}{E_{Pen}}$ no matter its changes come from the land surface

(changes in water availability) or the atmospheric aspects (the advections or the large-scale synoptic changes). From this point, we think the complementary principle could be further generalized to cover the later processes.

Although BC4 violates the assumptions of the CR, we don't agree that it makes the exception. When above processes are negligible, the coupled land-atmosphere system can be simplified as a one-dimension vertical column, in which the land surface and overlying atmosphere are fully connected. This condition can be satisfied more easily at monthly timescale than the daily timescale. For a natural landscape, advections from outside or large-scale synoptic changes would always exist, especially at a short timescale (daily for example). Besides, the relative importance of the two processes would vary with the wetness of the surface. Under water-limited conditions, the actual evaporation and potential evaporation are tightly linked via the surface, whereas the regional or large-scale advection plays a greater role than the landscape-scale processes under energy-limited conditions (Lintner et al., 2015). For a natural landscape, the simultaneous presence of above two processes may explain why the points of $\frac{E}{E_{Pen}}$ upon $\frac{E_{rad}}{E_{Pen}}$ are scatter. The further generalization of the complementary principle could help understand the variations of parameters of the complementary functions and aid in the parameter acquisition, thus enhancing the capability of the complementary principle to estimate evaporation.

3) BC4 and the third-stage of the relationship between $\frac{E}{E_{Pen}}$ and $\frac{E_{rad}}{E_{Pen}}$, which lead to a sigmoid function, are not well supported by empirical data.

Comment: We acknowledge that it is not easy to verify BC4, either to supply a visible example of the three-stage pattern (mainly the third stage) with observed data. According to BC4, $\frac{dy_H}{dx_H}$ should be zero at the maximum $x_H$, which implies that the slope should be near zero around the point with maximum $x_H$. By contrast, the corresponding boundary condition under the strict assumption of CR is $\frac{dy_H}{dx_H} = \alpha$ at the maximum $x_H$. Considering it is difficult to verify the derivative at a specific point, we evaluated the slopes of $y_H$ on a range of $x_H$ as a compromise. As the slope is not calculated at the specific point (maximum $x_H$), it will not be near zero. But the smaller

value of the slope for large values of $x_H$ compared to $\alpha$ can serve as an evidence. With a wide range of $x_H$ (larger than a critical value between 0.45~0.70) (HT18, Table 3), the calculated slopes ranged from 0.39- 1.30, and most of the sites (except for AU-How and NL-Loo) were characterized with the slopes smaller than $\alpha$. At two sites, the slopes are much less than 1. The calculated slopes would be much closer to zero if the evaluating was conducted much near to the maximum $x_H$.

We think that above empirical data would support that the growth of $y_H$ on $x_H$ has an upper flatness part. Based on the work of HT18, we have been working on the visible supports for the third-stage. In our preparing manuscript (Han et al., 2020), we found that the relationship between daily $\frac{E}{E_{Pen}}$ and $\frac{E_{rad}}{E_{Pen}}$ is characterized with an obvious upper flatness part (the third stage) over open water surface of lakes and ocean. The observed data at five lake sites from the Lake Taihu Eddy Flux Network (Lee et al., 2014) and the global ocean surface evaporation product (Version 3) from the OAFlux project (Yu & Weller, 2007) could serve as visible surports. The deviation of the Priestley-Taylor (PT) coefficient from a fixed value around 1.26 also indicates this upper flatness third stage.

Please refer to our preparing paper (Han et al., 2020) for details.

4) The derivations of BC4 and the sigmoid function were doubtful.

Comment: To the best of our knowledge, $x_m$ is a prognostic variable, which is calculated from the observed meteorological variables, and may be related to the aridity (Ma & Szilagyi, 2019). By contrast, in the SGC equation, $y_H = f(x_H, m, n, x_{min}, x_{max})$, we treated $x_H$ as the only independent variable, but the others as parameters. $x_{min}$ and $x_{max}$ are parameters, and would affect $y_H$ as parameters. From this point, $x_{min}$ is different from the variable $x_m$ in the "rescaled" CR.

We understand that the authors' concern comes from the considerations and treatments on the two parameters $x_{min}$ and $x_{max}$ during the derivation of BC4 and

the SGC equation. Following the method of the derivation of the complementary relationship by supposing a constant energy input (Bouchet, 1963; Brutsaert & Stricker, 1979), our derivation began with an assumption of certain magnitude of $E_{rad}$. The values of $x_{min}$ and $x_{max}$ depends on the relative magnitude of the maximum and minimum values of the aerodynamic term to $E_{rad}$. Thus, $x_{min}$ and $x_{max}$ would be roughly affected by the magnitude of $E_{rad}$. We clearly pointed out in HT18 that "$x_{max}$ is not independent of $E_{rad}$". Similar to the Priestley-Taylor coefficient, $x_{max}$ is thought to vary with the environment, but is used as a calibrated constant parameter in the SGC equation for convenience. $x_{max}$ is widely with a value of one.

Our derivation of the wet boundary conditions begins with an assumption of $E = E_{Pen} = E_{rad} + E_{aero}$, $\frac{E_{rad}}{E_{Pen}} = x_{max}$. Equation (17) in HT18 follows this assumption. There are two solutions for Equation (17) to hold. We adopts $\frac{dy}{dx}\Big|_{y=1} = 0$ in our SGC equation. However, $\frac{dy}{dx}\Big|_{y=1} = \frac{1}{x_{max}}$ is other solution, which is adopted in B2015.

To the best of our understanding, above two solutions represent two conditions, and can hold under each condition. $\frac{dy}{dx}\Big|_{y=1} = 0$ represents the condition when only advections or large-scale synoptic changes work, whereas the other represents only the land surface-atmosphere interactions work. The experimental studies in our preparing manuscript (Han et al., 2020) showed both the two conditions.

However, we have that the BCs under wet environments are complicated and are not well understood till now. This is why we stated at the end of HT20 that "it should be carefully examined for its physical base of the boundary conditions in a completely wet environment."

Other comments:

Line 30-39: The Priestley-Taylor line $E = \alpha E_{rad}$, which is equal to $\frac{E}{E_{Pen}} = \alpha \frac{E_{rad}}{E_{Pen}}$, is set as a limit on wet surface evaporation. Under the constraints on the parameters, the curve will not across the PT line. It is widely accepted that the Priestley-Taylor

coefficient varies with the environment. For a wet surface, we found that evaporation would be less than $E_{Pen}$, but roughly equal to $E_{PT} = \alpha E_{rad}$. The SGC equation can used to represent the wet surface evaporation with varying PT coefficient. The reference point, $(x_H = x_{max}, \ y_H = 1)$, represents the wet surface evaporation with the minimum Priestley-Taylor coefficient. This will be detailed in the manuscript in prepare (Han et al., 2020).

Line 54-56: We think that the wet surfaces with large-scale processes should not be considered as exceptional cases. Please refer to our preparing manuscript (Han et al., 2020).

Line 86: Because of the PT coefficient, $\frac{dy_H}{dx_B} = 1$ $(x_B = \frac{\alpha E_{rad}}{E_{Pen}})$ is equivalent to $\frac{dy_H}{dx_H} = \alpha$.

References:

Bouchet, R. (1963). Evapotranspiration réelle et potentielle, signification climatique. *International Association of Hydrological Sciences Publication, 62*, 134-142. Retrieved from http://12 - 12

Brutsaert, W. (1982). *Evaporation into the Atmosphere: Theory, History, and Applications*: D. Reidel-Kluwer, Hingham.

Brutsaert, W. (2015). A generalized complementary principle with physical constraints for land-surface evaporation. *Water Resources Research, 51*(10), 8087–8093, doi:8010.1002/2015WR017720. doi:10.1002/2015wr017720

Brutsaert, W., & Stricker, H. (1979). An advection-aridity approach to estimate actual regional evapotranspiration. *Water Resources Research, 15*(2), 443-450. Retrieved from http://16 - 16

Crago, R., & Qualls, R. (2018). Evaluation of the Generalized and Rescaled Complementary Evaporation Relationships. *Water Resources Research, 54*, 8086-8102. doi:10.1029/2018WR023401

Crago, R., Szilagyi, J., Qualls, R., & Huntington, J. (2016). Rescaling the complementary relationship for land surface evaporation. *Water Resources Research, 52*(11), 8461-8471. doi:doi:10.1002/2016WR019753

Han, S., Hu, H., & Tian, F. (2012). A nonlinear function approach for the normalized complementary relationship evaporation model. *Hydrological Processes, 26*(26), 3973-3981.

Han, S., & Tian, F. (2018a). Derivation of a sigmoid generalized complementary function for evaporation with physical constraints. *Water Resources Research, 54*(7), 5050-5068. doi:doi: 10.1029/2017WR021755

Han, S., & Tian, F. (2018b). Integration of Penman approach with complementary principle for evaporation research. *Hydrological Processes, 32*(19), 3051-3058.

Han, S., & Tian, F. (2019). Reply to Comment by J. Szilagyi and R. Crago on "Derivation of a sigmoid

generalized complementary function for evaporation with physical constraints". *Water Resources Research, 55*(2), 1734-1736. doi:10.1029/2018WR023844

Han, S., & Tian, F. (2020). A review of the complementary principle of evaporation: from the original linear relationship to generalized nonlinear functions. *Hydrology And Earth System Sciences, 24*(5), 2269-2285. doi:10.5194/hess-24-2269-2020

Han, S., Tian, F., Wang, W., & Wang, L. (2020). Sigmoid generalized complementary equation for evaporation over wet surfaces: as a nonlinear modification of the Priestley-Taylor equation.

Lee, X., Liu, S., Xiao, W., Wang, W., Gao, Z., Cao, C., . . . Wang, Y. (2014). The Taihu Eddy Flux Network: an observational program on energy, water and greenhouse gas fluxes of a large freshwater lake. *Bulletin Of The American Meteorological Society, 95*, 140530112919008.

Lintner, B., Gentine, P., Findell, K., & Salvucci, G. (2015). The Budyko and complementary relationships in an idealized model of large-scale land–atmosphere coupling. *Hydrology And Earth System Sciences, 19*(5), 2119-2131.

Liu, H., Blanken, P. D., Weidinger, T., Nordbo, A., & Vesala, T. (2011). Variability in cold front activities modulating cool-season evaporation from a southern inland water in the USA. *Environmental Research Letters, 6*(2), 024022. doi:10.1088/1748-9326/6/2/024022

Ma, N., & Szilagyi, J. (2019). The CR of Evaporation: A Calibration-Free Diagnostic and Benchmarking Tool for Large-Scale Terrestrial Evapotranspiration Modeling. *Water Resources Research, 55*(8), 7246-7274. doi:doi: 10.1029/2019wr024867

Monteith, J. L. (1965). *Evaporation and environment.* Paper presented at the Symposium of the Society of Experimental Biology, Cambridge.

Morton, F. I. (1983). Operational estimates of areal evapotranspiration and their significance to the science and practice of hydrology. *Journal Of Hydrology, 66*, 1-76. Retrieved from http://98 - 98

Shuttleworth, W. J. (1993). Evaporation. In *Handbook of Hydrology*. New York: McGraw-Hill.

Shuttleworth, W. J., Serrat-Capdevila, A., Roderickc, M. L., & Scottd, R. L. (2009). On the theory relating changes in area-average and pan evaporation. *Quarterly Journal Of The Royal Meteorological Society, 135*, 1230-1247.

Szilagyi, J., & Crago, R. (2019). Comment on "Derivation of a sigmoid generalized complementary function for evaporation with physical constraints" by S. Han and F. Tian. *Water Resources Research, 55*, 868–869.

Szilagyi, J., Crago, R., & Qualls, R. (2017). A calibration-free formulation of the complementary relationship of evaporation for continental-scale hydrology. *Journal of Geophysical Research: Atmospheres, 122*(1), 264–278.

Wang, L., Tian, F., Han, S., & Wei, Z. (2020). Determinants of the asymmetric parameter in the complementary principle of evaporation. *submitted to Water Resources Research*, 2019WR026570.

Yang, D., Sun, F., Liu, Z., Cong, Z., Ni, G., & Lei, Z. (2007). Analyzing spatial and temporal variability of annual water-energy balance in non-humid regions of China using the Budyko hypothesis. *Water Resources Research, 43*, W04426, doi:04410.01029/02006WR005224. Retrieved from http://161 - 161

Yu, L., & Weller, R. A. (2007). Objectively Analyzed Air–Sea Heat Fluxes for the Global Ice-Free Oceans (1981–2005). *Bulletin Of The American Meteorological Society, 88*(4), 527-540. doi:10.1175/bams-88-4-527

---

## Author Response (AR1)

Response to reviewer comments

We have already responded at length to the reviews by Han and Tian of the first draft of our
comment (that earlier response starts on the following page of this document). We are not going
to address all of the feedback provided in that review, but we do want to highlight some changes
we made in the manuscript in light of their review.

1. In the introduction, in response to the review by Han and Tian of the draft of this
comment, we removed the wording about different definitions of $\alpha$ and focused on the
incorporation of advective effects in the sigmoid function.
2. In section 2, we addressed the role of empirical versus physically-based models as well as
calibration. This topic was raised in the review by Han and Tian of the draft of this
comment, and we felt it was appropriate to address it here.
3. In section 3, we discussed the argument made in the review by Han and Tian of the draft
of this comment, namely that the flat part of the sigmoid curve only appears very near
$y_H=1$.
4. In section 5, we changed the wording regarding how "normal" it is for wet advection to
occur near $y_H=1$. At the end of the section we added two notes. First, an expression of the
desirability of handling advection in a CR formulation, and then a note that advection
plays an important role even for $y_H<1$.
5. In Section 6 we re-worded the summary of the argument in item 4 above.
6. We made multiple revisions to the reference citations.

Response to "Review of HESS-2020-310 'Comment on: A review of the complementary
principle of evaporation: From the original linear relationship to generalized nonlinear functions
by S. Han and F. Tian" (Reviews written by S. Han and F. Tian)

Richard D. Crago[1], Jozsef Szilagyi[2], Russell Qualls[3]

1 Department of Civil and Environmental Engineering, Bucknell University, Lewisburg, PA,
USA

2 Department of Hydraulic and Water Resources Engineering, Budapest University of
Technology and Economics, Budapest, Hungary; also at School of Natural Resources, University
of Nebraska, Lincoln, Lincoln, NE, USA

3 Department of Biological Engineering, University of Idaho, Moscow, ID USA

Introduction

We thank S. Han and F. Tian for their thoughtful review (hereafter, "HT2020b") of our comment
(hereafter, "CSQ2020") on Han and Tian (2020; hereafter "HT2020") and appreciate this
continued discussion of the complementary principle (CP). In CSQ2020, we agreed that the
Sigmoid Generalized Complementary (SGC) formulation is a serious development in CP
research that deserves careful consideration and analysis. However, we concluded that it was not
superior to other recent developments in the CP (e.g., Brutsaert, 2015; Crago et al. 2016; Crago
and Qualls, 2018; Szilagyi et al., 2017, Ma and Szilagyi, 2019). HT2020b was structured around
four claims, which we will discuss in order.

HT2020b Claim 1

HT2020b argue that two different approaches are both common and valuable in hydrology
research. The first consists primarily of "calibrating parameters for the fitting of observed points
and proposing a method to determine the parameters *in priori*." The second consists primarily of
developing "approaches…carefully conducted on a physical basis." We agree--methods that
consistently and accurately reproduce measurements are the most valuable. However, we find the
second type of models to be more likely to generalize well and to apply well outside the
validation range. We also acknowledge the reviewers' efforts as much as possible to ground their
own research on a physical basis. We agree both methodologies should be explored, but would
much prefer to proceed with physically-based approaches when possible.

HT2020b claim 2

Second, HT2020b address interpretation of the CP in conditions where large-scale advection or
entrainment of free-atmosphere air partially disconnect the atmospheric boundary layer (ABL)
from the condition of the surface. CSQ2020 argued that the CP is no longer valid under these
conditions. That is, the logic of the CP requires that the ground and ABL are connected, so that
the condition (temperature, humidity, wind speed, etc.) of the atmosphere is adjusted to the condition of the surface, particularly the availability of moisture at the surface. We agree that it is
possible, in principle, to extend a method originally formulated as a CP equation so that it applies
under conditions dominated by these large-scale conditions. Han and Tian (2018; hereafter
"HT2018") attempt to do this by arguing that over wet surfaces actual regional evaporation $E$
and Penman evaporation $E_{pen}$ are nearly identical so that if $E_{pen}$ is increased by dry advection, $E$
would increase at essentially the same rate.  We agreed in our comment that this is possible, but
that it implies that the CP is invalid because the conditions in the ABL are disconnected from
those at the surface. This brings the argument back to claim 1, because if the SGC works under
these conditions, it is not because it captures the physical processes, but because it successfully
matches the data.

HT2020b claim 3

In HT2020 and HT2018, experimental data from around the world are presented to demonstrate
the existence of the three-stage pattern they advocate. CSQ2020 noted that other formulations,
such as that of Brutsaert (2015) could also be said to have three comparable phases, and that the
claim to have a horizontal upper (wet-surface) limit to the third stage is not supported by these
data. HT2020b responded that the flat portion (derivative of zero) only strictly applies at a single
point on the curve, so that graphs of data points would not necessarily reveal the flatness of the
curve. This is a perfectly logical argument, but it means that the primary evidence for a proposed
flat third-stage is not empirical but theoretical.

HT claim 4

The most powerful theoretical defense of the flat third stage of the SGP is found in HT2018, in
which they derive slopes for the SGP curve at $x_{min}$ and $x_{max}$, the dry and wet limits, respectively.
HT2020b wrote that the SGC equation can be expressed $E/E_{pen}=f(E_{rad}/E_{pen}, m, n, x_{min}, x_{max})$,
where $E_{rad}$ is the first term of $E_{pen}$. But HT2020b stated that, in HT2018, $E_{rad}/E_{pen}$ was treated as
the only independent variable, with the others as parameters. HT2018 and HT2020b were not
obligated to include $x_{min}$ as an important variable that can be calculated independently for each
data point as proposed in our papers (Crago et al. 2016; Crago and Qualls, 2018; Szilagyi et al.,
2017, Ma and Szilagyi, 2019). However, CSQ2020 noted that the assumption that $E_{rad}/E_{pen}$ was
the only variable in $f$ ruled out any version of our "rescaled" CP formulation. Incorporation of
this variable $x_{min}$ into the CP actually changes the functional form of the CP, which presumably
could change the slope, particularly at the lower limit.

The first step in the derivation by HT2018 (after defining $E/E_{pen}$ as a function of $E_{rad}/E_{pen}$ only)
was to take partial derivatives of $E$ with respect to $E_{rad}$ and $E_{aero}$ (i.e., the second term of $E_{pen}$),
resulting in equation (17) of HT2018. CSQ2020 found this problematic because the process did
not consider $x_{max}$ (or $x_{min}$, but we will focus on $x_{max}$ in this paragraph) to be a variable in this
process. The partial derivatives would have involved more terms, such as $(\partial E/\partial x_{max})(\partial x_{max}\partial E_{rad})$
which would not be easy to analyze. Treating $x_{max}$ as only a parameter resulted in (17). But later
in the derivation, HT2018 claimed that $\partial x_{max}/\partial E_{rad}$ is not zero; this claim led directly to the flat
third stage of the SGC curve. But CSQ2020 noted that, if $x_{max}$ is a constant or parameter, this derivative must be zero. HT2020b responded that $x_{max}$ was in fact treated as a parameter, not a
variable, but also that "$x_{max}$ is thought to vary with the environment," and "$x_{max}$ is not
independent of $E_{rad}$." These quotes seem to support the critique of CSQ2020 that $x_{max}$ is treated
as both a constant and as a variable in the same derivation. If $\partial x_{max}/\partial E_{rad}$ is not zero, then $x_{max}$
must be treated as a variable when the partial derivatives are taken in the first step of the
derivation.

To their credit, HT2020b do acknowledge that the limits to the CP are not well understood. Their
surmise that this is due to the relative roles of advection and surface wetness at $x_{max}$ seems
plausible.

Summary

The CP is a fascinating concept. The principle can be stated in one or two sentences and in
equations with only a few variables, but the application of the principle and interpretation of the
variables is surprisingly complicated and some of the concepts are elusive. We have learned a
great deal in thinking through the issues raised by these authors. We find at the end of this
process that there are significant areas of agreement between us and HS2020b, and decreasing
areas of disagreement. Specifically, we agree that both largely empirical and process-based
approaches are valuable, and that large-scale advection must have an impact on the CP. But,
while we appreciate the contributions of S. Han and F. Tian to this research, we still do not find
arguments for the SGC formulation of the CP to be convincing.

[revised manuscript text omitted]